# TP53INP2 Promotes Bovine Adipocytes Differentiation Through Autophagy Activation

**DOI:** 10.3390/ani9121060

**Published:** 2019-12-02

**Authors:** Weiyi Zhang, Peiwei Li, Shijie Wang, Gong Cheng, Li Wang, Xue Mi, Xiaotong Su, Yaning Wang, Linsen Zan

**Affiliations:** 1College of Animal Science and Technology, Northwest A&F University, Yangling 712100, Shaanxi, Chinachenggong@nwafu.edu.cn (G.C.); lw@nwafu.edu.cn (L.W.); rsmx@nwafu.edu.cn (X.M.); xiaotongsu@gmail.com (X.S.); wangyn1992@outlook.com (Y.W.); 2National Beef Cattle Improvement Center in China, Yangling 712100, Shaanxi, China

**Keywords:** TP53INP2, autophagy, adipocyte differentiation, lipid droplets, PPARγ

## Abstract

**Simple Summary:**

In this article explore the role of the bovine *TP53INP2* gene in adipocyte differentiation and its function in autophagy during the early stage of adipocyte differentiation. In our work we found that a novel, important autophagy related protein TP53INP2 can activate autophagy during the early stage of differentiation in bovine adipocytes and positively regulate adipocyte differentiation by affecting autophagy. Furthermore, we demonstrated that peroxisome proliferator-activated receptor gamma (PPARγ) also contributed to the function of TP53INP2 in modulating adipocyte differentiation. The study of the function of bovine *TP53INP2* gene on adipocyte differentiation has not been reported, therefore, we have decided to focus on Qinchuan cattle, one of the five important cattle breeds in China. We propose that the *TP53INP2* gene may affect the meat quality of Qinchuan cattle by regulating lipid deposition, and may shed new light on the developmental mechanisms of adipose development.

**Abstract:**

Tumor protein p53 inducible nuclear protein 2 (TP53INP2) is a key positive regulator of autophagy, and it has been shown to modulate adipocyte differentiation. However, the molecular mechanism involved in autophagy regulation during adipocyte differentiation has not been clarified. Our experiments were intended to investigate whether TP53INP2 is involved in the regulation of autophagy during bovine adipocyte differentiation and how TP53INP2 affects the differentiation of bovine adipocytes. In our research, using RT-qPCR and Western blot methods, we found that the overexpression of TP53INP2 resulted in the upregulation of adipogenesis and autophagy-related genes, and autophagy flux and the degree of differentiation were detected by LipidTOX™ Deep Red Neutral Lipid staining and dansylcadaverine staining, respectively. The knockdown of TP53INP2 produced results that were the inverse of those produced by the overexpression of TP53INP2. Overall, our results suggested that TP53INP2 can activate autophagy during the early stage of differentiation in bovine adipocytes and positively regulate adipocyte differentiation by affecting autophagy. Additionally, peroxisome proliferator-activated receptor gamma (PPARγ) also contributed to the function of TP53INP2 in modulating adipocyte differentiation.

## 1. Introduction

Autophagy is a lysosomal degradation pathway [1] that can remove excess or damaged organelles or macromolecules to maintain the recycling of intracellular substances [1,2,3] and the homeostasis of the internal environment [4]. Autophagy also plays an important role in some diseases [5]. It is initiated in a double-membrane structure called the autophagosome after membrane expansion. Then, the whole membrane encloses the contents. Finally, autophagosomes will move toward and fuse with lysosomes and assemble into autolysosomes that contain hydrolases that can degrade substances [6,7]. Many studies have confirmed that autophagy is involved in the differentiation of adipocytes and plays an essential role in the early stage of differentiation [8,9,10]. Inhibiting autophagy during the process of preadipocyte differentiation into mature adipocytes blocks normal differentiation. The characteristics of brown adipocytes appear in white adipocytes when autophagy is blocked in adipocyte differentiation; lipid droplets become smaller, the number of mitochondria increases, the β-oxidation rate and insulin sensitivity increase, and the lipid content is reduced [11,12]. The tissue-specific knockout of the autophagy-related gene *ATG5* in mice will affect adipogenesis and lipid accumulation [13]. ATG7 encodes an E1-like activating enzyme that is necessary for autophagy. A previous study revealed that ATG7 could regulate adipocyte differentiation in mice [14].

Tumor protein p53 inducible nuclear protein 2 (TP53INP2), also known as DOR or PINH, is a dual regulator of transcription and autophagy. The tumor protein p53 inducible nuclear protein family has two members, TP53INP1 and TP53INP2. Extensive research has been conducted on TP53INP1, which is involved in the cell stress response, inhibits cell proliferation and promotes apoptosis. TP53INP2 is a vital paralog of TP53INP1, which is a positive regulator of autophagy. However, there has been less research on TP53INP2. Existing reports have shown that TP53INP2 can interact physically with some important autophagy-related genes, such as *MAP1LC3A*, *GABARAP* and *GABARAPL2*, and it can recruit them to autophagosome membranes and link them to *VMP1*. All of the autophagy-related genes mentioned above play pivotal roles in autophagosome development. Recently, TP53INP2 has been found to play a novel role in death-receptor signaling, and it can positively regulate apoptosis [15,16,17]. The latest results have reported that TP53INP2 could negatively regulate adipogenesis in human and mouse preadipocytes [18].

Adipose tissue is a vital endocrine organ that maintains energy homeostasis and is critical for mammalian development [19,20,21]. It has been demonstrated that adipose tissue could regulate nutritional metabolism and lipid storage [22,23]. Adipogenesis is a complex process, and differentiation is a critical process, after which adipocytes are occupied by large lipid droplets rich in TG [24,25]. Many studies have reported that several important transcription factors, such as peroxisome proliferator-activated receptors (PPARγ), CCAAT/enhancer-binding protein alpha (CEBPα), and fatty acid-binding protein (FABP4) regulate differentiation [26,27,28]. In addition to investigating these regulators, it is also necessary to study the mechanisms of the regulation of differentiation involving other novel genes.

To date, there have been few reports on autophagy regulating the differentiation of bovine adipocytes, and the study of the bovine *TP53INP2* gene has not been reported. Therefore, we have decided to focus on Qinchuan cattle, one of the five important cattle breeds in China. We propose that the *TP53INP2* gene may affect the meat quality of Qinchuan cattle. Therefore, we explored the role of the bovine *TP53INP2* gene in adipocyte differentiation and its function in autophagy during the early stage of adipocyte differentiation. Ultimately, our experiments found that PPARγ and autophagy contributed to the function of TP53INP2 in bovine adipocyte differentiation. Therefore, we suggest that TP53INP2 is the key regulator of differentiation in bovine adipocytes.

## 2. Materials and Methods

Bovine pre-adipocytes were isolated from a one-day-old healthy calf born at the experimental base of the National Beef Cattle Improvement Center (Yangling, China). The care and feeding of the animals used in this study were approved by the Institutional Animal Care and Use Committee of China (College of Animal Science and Technology, Northwest A&F University, China; No. 2013-23, 20 April 2013). The implementation of the animal experimental procedures was performed in strict accordance with the guidelines of the Administration of Affairs Concerning Experimental Animals (Ministry of Science and Technology, China, 2004).

### 2.1. Isolation of Bovine Adipocytes

Subcutaneous adipose tissue was harvested under sterile conditions and placed in 1x PBS (Gibco, Grand Island, NY, USA) with 10% penicillin/streptomycin (Gibco). It was then immediately taken out. Under a stereo-dissecting microscope, the subcutaneous adipose tissues were dissected away from the connective tissue and blood vessels with sterile forceps and scissors. The adipose tissue was cut into small pieces. The minced tissue blocks were digested with 0.25% collagenase I (Sigma, Kawasaki City, Japan) and 0.1% dispase II (Roche, Basel, Switzerland) for 1–2 h at 37 °C in a water bath. The digested mixture was filtered through an 80-μm cell strainer and centrifuged at 1500*g* for 10 min, and the supernatant was discarded. The cells were resuspended in complete growth medium (Dulbecco’s modified Eagle medium/F-12 (DMEM/F-12), Gibco) with 15% fetal bovine serum (Gibco) and 1% penicillin/streptomycin) and seeded in 60 mm Petri dishes, and the medium was changed every two days.

### 2.2. Cell Culture and Transfection

Bovine pre-adipocytes were cultured in DMEM/F-12 with 15% fetal bovine serum (Gibco) and 1% penicillin and streptomycin at 5% CO_2_ and 37 °C. Adipocyte differentiation was induced by an induction medium containing DMEM/F-12 with 15% FBS, 1% penicillin and streptomycin, 0.50 mM isobutylmethylxanthine (IBMX, Sigma, Kawasaki City, Japan), 1 mM Dexamethasone (Dex, Sigma, Saint Louis, MO, USA), and 1 mg/mL insulin. The first and second days of treatment with the induced differentiation medium were referred to as D0 and D1. Then, the maintenance medium (DMEM/F-12 with 15% FBS, 1% penicillin and 1 mg/mL insulin) was changed every other day until the fifth day. The cells were transfected when the density reached approximately 70% and were starved for two hours in serum-free OptiMEM (Gibco) before transfection. The CDS sequence of TP53INP2 cDNA was subcloned into the pcDNA3.1(+) vector, which was used to overexpress TP53INP2, and was transfected, along with pcDNA3.1(+) empty vector, siRNA (20 μM), and NC (negative control, 20 μM) (Ribobio, Guangzhou, China) into cells with the FuGENE^®^ HD transfection reagent according to the transfection protocol. Rosiglitazone (2 μM) was used to increase the expression level of PPARγ.

### 2.3. Quantitative RT-PCR

Total cellular RNA was extracted with RNAiso Plus (Takara, Mountain View, CA, USA) reagent according to the manufacturer’s instructions. cDNA synthesis was performed using an RT reagent kit with gDNA Eraser (Takara) according to general experimental procedures. The relative expression of mRNA was determined by quantitative RT-PCR using the TB Green Premix Ex Ta II kit (Takara). The results of RT-PCR were calculated by using the 2*^−ΔΔCt^* method [29]. β-actin was used to normalize all genes expression levels. All primers were synthesized by TSINGKE Biological Technology Co. (Xi’an, Shanxi, China), and the sequence information is listed in Table 1.

### 2.4. Western Blot

Total cellular protein was isolated from adipocytes with a protein extraction kit (Solarbio Company, Beijing, China) after mixing with protein loading buffer and denaturation for 10 min in a 100 °C metal bath. A 20-µg protein sample was electrophoresed on a 12% gel, and the protein was transferred to a PVDF membrane after gel electrophoresis. Next, the membrane was incubated with antibodies against β-ACTIN (1:5000, NOVUS, HK, NP_776404.2), TP53INP2 (1:2000, AVIVA Systems Biology, San Diego, CA, USA, XP_003586891.1), PPARγ (1:1000, Boster, Wuhan, China, NP_851367.1), PLIN2 (1:2000, Abcam, Cambridge, UK, NP_776405.1), FASN (1:2000, Abcam, NP_777087.1), p62 (1:2000, Abcam, NP_788814.1), or LC3 (1:2000, Abcam, NP_001001169.1) for 12 h at 4 °C. After washing the membrane three times (10 min each) with PBS-Tween 20, the membrane was incubated with the secondary antibody for 1 h and then washed three times with PBS-Tween 20 (10 min each). Finally, equal amounts of luminol reagent and peroxide solution were mixed in an EP tube and added dropwise to the PVDF membrane. The Gel Doc™ XR+ Gel Documentation System (Bio-Rad, Hercules, CA, USA) was used to detect the immunoreactivity.

### 2.5. Dansylcadaverine Staining and Lipid Droplet Staining

Dansylcadaverine, a fluorescent stain, is commonly used to monitor autophagy vacuoles. Adipocyte lipid droplet staining was performed with HCS LipidTOX™ Deep Red Neutral Lipid Stain (Thermo Fisher, Waltham, MA, USA). After washing cells with PBS three times, the adipocytes were fixed with 4% formaldehyde and incubated for 30 min at room temperature. Next, the formaldehyde was removed, and the cells were gently washed with PBS three times, after which 0.5 mM dansylcadaverine stain (Sigma) was added (1 uL per well in a 6-well plate) and incubated at 37 °C for 40 min. After that, the dansylcadaverine stain was removed, and the cells were washed with PBS 3 times. LipidTOX™ neutral lipid stain was added (1 uL per well in a 6-well plate), and the cells were incubated at room temperature for 30 min. Finally, the LipidTOX™ stain was removed, and the cells were washed three times with PBS. Then, DAPI stain (Sigma) was added (1 uL per well in a 6-well plate) and incubated at room temperature for 10 min before cell imaging.

### 2.6. Statistical Analysis

All data in this study are represented as the mean ± SEM and were analyzed with GraphPad Prism 5.0 (software Inc., San Diego, CA, USA). The significance of the differences between groups was calculated by a *t*-test or ANOVA (*, *p* < 0.05; **, *p* < 0.01).

## 3. Results

### 3.1. Constructing and Transfecting pcDNA3.1(+)-TP53INP2 to Overexpress TP53INP2 and Transfecting Si-TP53INP2 to Interfere with TP53INP2

The complete CDS sequence of the bovine *TP53INP2* gene was subcloned into the pcDNA3.1(+) vector. When the cell density reached 70%, the overexpression vector was transfected into adipocytes. At 48 h after transfection, we tested the overexpression efficiency of TP53INP2, and the mRNA level of *TP53INP2* was upregulated nearly 500-fold compared to that in the negative control cells (Figure 1a). The TP53INP2 protein level was also significantly increased (Figure 1b, Appendix A). In addition, we synthesized and transfected three small interfering RNAs targeting the bovine *TP53INP2* gene. Similarly, the interference efficiency was assessed at 48 h after transfection, and the siRNA with the highest interference efficiency was selected. RT-qPCR analysis showed that the mRNA expression of *TP53INP2* was decreased nearly 80% compared to that in the NC (Figure 1c). Western blot analysis showed that the protein level was also significantly decreased (Figure 1d, Appendix A). In addition, we also detected overexpression and interference efficiency of mRNA levels at 24 h, 72 h and 150 h after transfection (Appendix A).

### 3.2. TP53INP2 Promotes Differentiation in Bovine Adipocytes

We first determined the mRNA expression of *TP53INP2* during differentiation (Figure 2a). To explore the role of TP53INP2 in adipocyte differentiation and adipogenesis, we transfected adipocytes with siRNA (small interfering RNA) and NC or NC alone as a control. Our results showed that the mRNA and protein levels of PPARγ and perilipin-2 (PLIN2) gradually increased during differentiation, while the level of fatty acid synthase (FASN) increased first and then decreased. The knockdown results showed that both the mRNA and protein expression of the differentiation marker PPARγ and the adipogenesis markers PLIN2 and FASN were significantly decreased by knocking down TP53INP2 during differentiation (Figure 2b–e, Appendix A). In addition, LipidTOX™ Deep Red Neutral Lipid Stain showed that the lipid droplets in cells in the si-TP53INP2 group were clearly less than those in the control group (Figure 3a, Appendix A). To further explore the function of TP53INP2 in adipocytes, we also transfected a pcDNA3.1(+)-TP53INP2 plasmid to overexpress TP53INP2 and an empty vector as a negative control. The overexpression results showed inverse trends; both the mRNA and the protein expression of the differentiation marker PPARγ and the adipogenesis markers PLIN2 and FASN were significantly increased during differentiation (Figure 2f–i, Appendix A). Consistent with this, the overexpression of TP53INP2 increased the lipid droplet content (Figure 3b, Appendix A). In general, our experimental results showed that TP53INP2 promoted bovine adipocyte differentiation.

### 3.3. TP53INP2 Induces Autophagy During the Differentiation

We also evaluated the role of TP53INP2 in autophagy during the differentiation of bovine adipocytes. In our results, the mRNA levels of *ATG7* and *BECN1* were gradually upregulated during differentiation, and the LC3 protein level was also upregulated. Knocking down TP53INP2 during differentiation could reduce the mRNA levels of autophagy marker genes, including *ATG7* and *BECN1*. The protein level of LC3-II was decreased, and the protein level of p62 was increased (Figure 4a–c, Appendix A). To further confirm the role of TP53INP2 in autophagy, we observed autophagy via dansylcadaverine staining. The staining results showed that knocking down TP53INP2 significantly decreased autophagy when compared with the negative control. At the same time, when autophagy was inhibited, it could be observed that the content of the lipid droplets was also reduced (Figure 5a, Appendix A). The overexpression of TP53INP2 resulted in an inverse tendency, and the expression of *ATG7*, *BECN1* and LC3-II was increased and that of p62 was decreased (Figure 4d–f, Appendix A). Furthermore, the staining assay revealed that autophagy was induced. (Figure 5b, Appendix A). Overall, TP53INP2 positively regulated autophagy during the differentiation of bovine adipocytes.

Our experimental data demonstrated that TP53INP2 could regulate bovine adipocyte differentiation; therefore, altering the level of autophagy via changing the expression level of TP53INP2 affected the differentiation of bovine adipocytes.

### 3.4. TP53INP2 Promotes the Adipocytes Differentiation Through PPARγ the Pathway

According to the experimental results, we found that when the expression level of TP53INP2 was overexpressed or reduced, the protein and mRNA levels of PPARγ, FASN, and PLIN2 were significantly increased or decreased; however, PPARγ is a crucial transcription factor that can regulate the expression of other genes. Therefore, we hypothesized that TP53INP2 may regulate bovine adipocyte differentiation through the PPARγ pathway. To explore the mechanism by which TP53INP2 regulates adipocyte differentiation, we performed a rescue experiment on the third day after the induction of differentiation. The results indicated that the overexpression of PPARγ induced by the PPARγ agonist rosiglitazone restored the reduction of lipids caused by decreasing the expression of TP53INP2 (Figure 6e, Appendix A) and upregulated the protein and mRNA levels of the differentiation marker PLIN2 and the protein level of FASN (Figure 6a–d, Appendix A). Hence, these results implied that TP53INP2-regulated adipocyte differentiation may be related to PPARγ.

## 4. Discussion

According to the expression of *TP53INP2*, we found that the peak in mRNA expression on the second day after the induction of differentiation was significantly higher than that on the uninduced day and the fourth day after induction. Some studies have confirmed that autophagy plays an essential role in the early stage of adipocyte differentiation [30]. Therefore, we chose to examine cells transfected with the overexpression vector, siRNA, empty vector and NC (control) on D0 (uninduced), D1, D3, and D5 after differentiation induction. A previous study demonstrated that after knocking down TP53INP2 in 3T3-L1 cells, adipocytes became larger and contained aggregated triglycerides. In addition, the inhibition of TP53INP2 expression during 3T3-L1 preadipocyte differentiation significantly increased the expression of the differentiation markers PPARγ and CEBPα. Perlipin A/B and GLUT4 were also upregulated. In contrast, the overexpression of TP53INP2 in adipocytes decreased triglyceride levels, and PPARγ and CEBPα expression were also downregulated. In conclusion, TP53INP2 is a negative regulator of 3T3-L1 preadipocyte differentiation, which also suggests that TP53INP2 participates in the regulation of adipocyte differentiation [18]. Our results indicated that the overexpression of TP53INP2 promoted bovine adipocyte differentiation through increasing the amount of lipids and upregulating PPARγ, PLIN2, and FASN at both the mRNA and protein levels. A reduction in TP53INP2 resulted in the inverse of that observed in the overexpression group. The most important physiological function of PPARγ is to positively regulate adipogenesis [31]. In our study, PPARγ expression was gradually increased after inducing differentiation, and both the mRNA and protein levels of PPARγ were positively correlated with the expression of TP53INP2 during differentiation. PLIN2 is a lipid-droplet surface membrane protein that localizes to the surfaces of lipid droplets. PLIN2 promotes the formation of early lipid droplets in bovine adipocytes [32]. Our work indicated that TP53INP2 and PLIN2 are positively correlated, indicating that TP53INP2 participates in the regulation of early lipid droplet formation. FASN is a multifunctional enzyme, and its main function is to promote the synthesis of long-chain fatty acids [33]. Our research showed that FASN began to be expressed after the induction of differentiation and that TP53INP2 could significantly change the expression of FASN, which suggested that TP53INP2 is involved in lipid accumulation. Therefore, we hypothesize that the reasons for the different effects of TP53INP2 in mouse and bovine adipocytes are that (a) the *TP53INP2* gene has species-specific differences in mice and cows individuals and (b) the inhibition of TP53INP2 expression in 3T3-L1 preadipocytes was not associated with decreased autophagy flux [18]; however, in our study, the suppression of TP53INP2 expression in bovine adipocytes significantly inhibited autophagy. Hence, we speculated that TP53INP2 promoted differentiation by affecting autophagy and (c) our results showed that TP53INP2 activated the transcriptional activity of PPARγ, therefore, we hypothesized that TP53INP2 may promote bovine adipocyte differentiation through the PPARγ pathway. However, the specific mechanism is still unclear, and we need to further confirm this.

Some reports have demonstrated that TP53INP2 is essential for autophagy in mammalian cells [34]. It can shuttle between the nucleus and cytoplasm. When autophagy occurs, TP53INP2 relocates to the autophagosome and interacts with the autophagy-associated proteins GATE16 and LC3 [17]. In addition, it has been reported that the inhibition of TP53INP2 expression by small interfering RNA can repress autophagosome formation. Autophagy flux is very low in 3T3-L1 preadipocytes, and the knockdown of TP53INP2 during adipocyte differentiation has no effect on autophagy [18]. However, the present study demonstrated that TP53INP2 promoted autophagy in bovine adipocytes. ATG7 encodes an E1-like activating enzyme that is necessary for autophagy. A previous report revealed that ATG7 regulates adipocyte differentiation in mice [14]. *BECN1* is also an autophagy marker gene. Our research showed that the mRNA expression levels of *ATG7* and *BECN1* gradually increased during adipocyte differentiation, and the overexpression or deficiency of TP53INP2 significantly upregulated or downregulated the *ATG7* and *BECN1* mRNA levels, respectively. We also measured the protein expression levels of LC3 and p62, which are two marker proteins of autophagy. Some reports have indicated that LC3 protein translation produces two forms, LC3-I and LC3-II. When autophagy is activated, LC3-I transforms into LC3-II, and the amount of LC3-II is related to the level of autophagy [35]. In our study, interference with TP53INP2 expression, which suppressed the induction of differentiation, caused LC3-I to hardly be expressed in the si-TP53INP2 group. After the induction of differentiation, LC3-I was expressed until the third day after the induction of differentiation; LC3-I conversion into LC3-II and the ratio of LC3-II to LC3-I in the si-TP53INP2 group was always lower than that in the control group during differentiation. However, after transfection of the overexpression vector, LC3-I began to be converted into LC3-II in undifferentiated adipocytes. As differentiation progressed, the ratio of LC3-II to LC3-I gradually increased, and the ratio in the overexpression group was always higher than that in the empty vector group. p62 can also be used to monitor autophagy flux. When autophagy occurs, p62 binds to LC3 and is efficiently degraded by autophagy [36]. In our research, due to the deficiency of TP53INP2, p62 could not be degraded. In the control group, in which differentiation was induced on the third and fifth days, p62 was almost completely degraded, and the protein expression level of p62 in the si-TP53INP2 group was also significantly lower on the third and fifth days than on the other differentiation days. After the induction of differentiation on the third day, p62 was hardly expressed in the overexpression group. The above experimental results show that TP53INP2 could regulate autophagy during the early stage of differentiation in bovine adipocytes.

Many reports have shown that autophagy could regulate the differentiation of adipocytes, especially in the early stage of adipocyte differentiation. Autophagy plays a very important role [37,38]. Specifically, by knocking out ATG7 in adipose tissue, mutant mice become very slim and have less white adipose tissue content than wild-type mice; the number of adipocytes in mutant mice are decreased, and the mitochondrial content is increased [14]. The specific knockdown of ATG7 in mouse 3T3-L1 preadipocytes led to reduced TG deposition and the downregulation of marker genes of adipocyte differentiation [39]. Similarly, the specific knockdown of ATG5, which encodes a protein necessary for autophagy in mouse MEFs, results in a decrease in normal differentiation [13]. The regulation of the differentiation of bovine adipocytes by autophagy has not been reported. In this study, the measurement of *TP53INP2* mRNA expression during differentiation showed a high abundance of TP53INP2 on the second day after the induction of differentiation, and TP53INP2 activated autophagy after the induction of differentiation, which is consistent with studies that showed that autophagy played an important role in the 2nd and 3rd days of adipocyte differentiation. In addition, our experimental results proved that TP53INP2 has a significant effect on the early differentiation of bovine adipocytes. Therefore, we speculated that TP53INP2 may regulate the differentiation of bovine adipocytes by affecting autophagy. In addition to functioning as an autophagy-related protein, TP53INP2 is also an important transcription factor. A previous study found that TP53INP2 negatively regulates adipocyte differentiation in 3T3-L1 cells by altering the expression of PPARγ and CEBPα [18]. PPARγ is a very crucial transcription factor and is highly expressed in white and brown adipose tissue. It has also been demonstrated that PPARγ plays a vital role in the formation of mature adipocytes [40]. Therefore, to investigate whether TP53INP2 can regulate adipocyte differentiation through PPARγ in bovine adipocytes, we performed a rescue experiment with rosiglitazone, a PPARγ agonist [41]. The results showed that the activation of PPARγ activity could significantly compensate for the decrease in lipid droplets and the upregulation of PLIN2 and FASN caused by the knockdown of TP53INP2. We hypothesized that PPARγ may contribute to the function of TP53INP2 in bovine adipocytes.

Several studies have demonstrated that mTOR regulates the synthesis of proteins and leptin in adipocytes. It is also documented that mTOR regulates the activity of PPARγ, so mTOR is critical for the initiation of adipocyte differentiation and the maintenance of lipid synthesis [42]. The mTOR pathway has also been reported to have important regulatory effects on autophagy [43]. Thus, further research should focus on the upstream signaling pathways associated with autophagy and differentiation to help improve the understanding of the mechanisms underlying the regulation of adipocyte differentiation by autophagy.

In summary, we found that a novel, important autophagy protein could regulate differentiation in bovine adipocytes. TP53INP2 induced autophagy and PPARγ expression, which may be a key factor in adipocyte differentiation.

## 5. Conclusions

In conclusion, our study found an autophagy-related protein that could promote differentiation in bovine adipocytes; in addition, it could regulate autophagy during the early stage of differentiation in bovine adipocytes. Therefore, we hypothesized that TP53INP2 regulated the differentiation of bovine adipocytes by altering the level of autophagy. Finally, we hypothesized that PPARγ may contribute to the function of TP53INP2 in bovine adipocytes via the rescue experiment.

## Figures and Tables

**Figure 1 animals-09-01060-f001:**
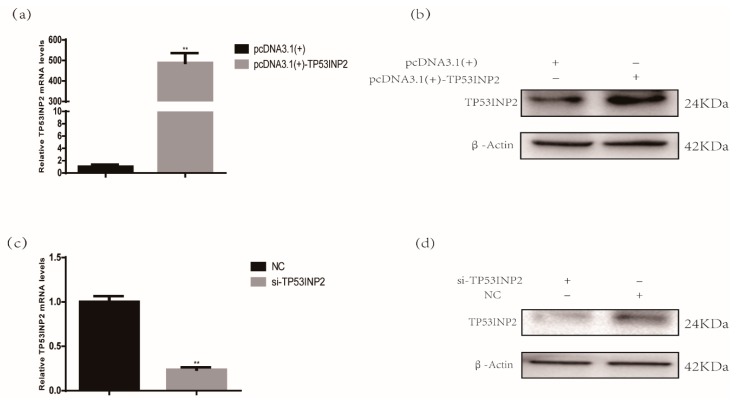
Detection of overexpressing and interference efficiency. (**a**) The mRNA and (**b**) protein levels of overexpressing efficiency after transfection for 48 h were nearly 500-fold and 2-fold, respectively. (**c**) The mRNA and (**d**) protein levels of interfering efficiency after transfecting for 48 h were nearly 80%. Bar graphs are expressed as mean ± S. E.M. (*n* = 3), **, *p* < 0.01, compared with NC and empty vector. NC, negative control.

**Figure 2 animals-09-01060-f002:**
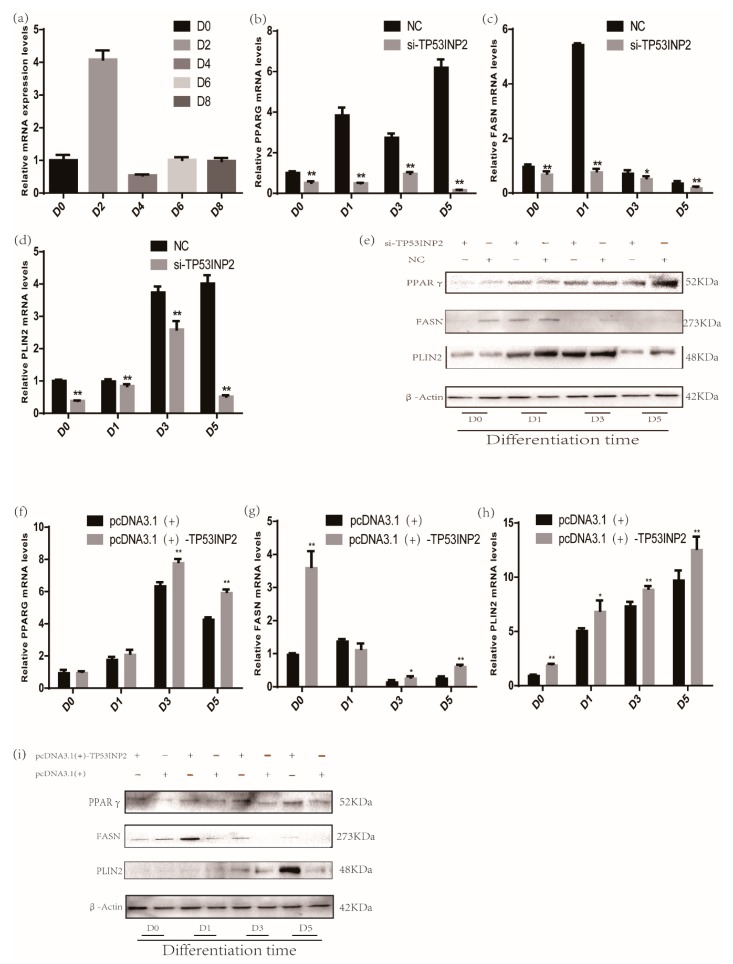
Effect of TP53INP2 on the differentiation of bovine adipocytes. (**a**) The mRNA expression pattern of *TP53INP2* during differentiation was measured by RT-qPCR. (**b**) *PPARγ*, (**c**) *PLIN2* and (**d**) *FASN* mRNA levels decreased during differentiation and were determined by RT-qPCR after knocking down of TP53INP2. (**e**)The protein expression levels of PPARγ, PLIN2 and FASN decreased during differentiation and were measured by Western blot after knocking down of TP53INP2. (**f**) *PPARγ*, (**g**) *PLIN2* and (**h**) *FASN* mRNA levels increased during differentiation and were determined by RT-qPCR after overexpressing of TP53INP2. (**i**) The protein expression levels of PPARγ, PLIN2 and FASN increased during differentiation and were measured by Western blot after overexpressing of TP53INP2. Bar graphs are expressed as mean ± S. E.M. (*n* = 3), *, *p* < 0.05; **, *p* < 0.01, compared with NC and empty vector. NC, negative control.

**Figure 3 animals-09-01060-f003:**
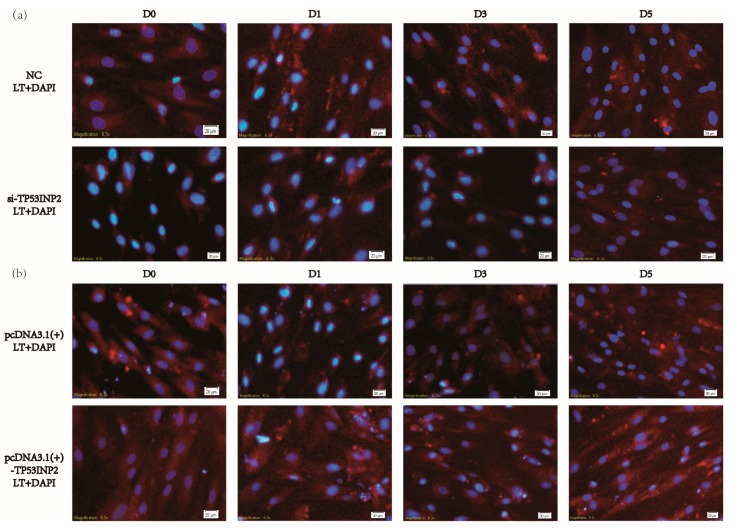
Lipid staining of lipid droplets in bovine adipocytes during differentiation. (**a**) Lipid staining of differentiated adipocytes by HCS LipidTOX™ Deep Red Neutral Lipid Stain (red) and nucleus were stained via DAPI (blue), the lipid droplets were less than the control group after knocking down of TP53INP2 (20 μm). (**b**) Lipid staining of differentiated adipocytes via HCS LipidTOX™ Deep Red Neutral Lipid Stain (red) and nucleus were stained via DAPI (blue), the lipid droplets were more than the control group after overexpressing of TP53INP2 (20 μm). Compared with NC and empty vector. NC, negative control; LT, HCS LipidTOX™ Deep Red Neutral Lipid Stain; DAPI, 4’,6-diamidino-2-phenylindole.

**Figure 4 animals-09-01060-f004:**
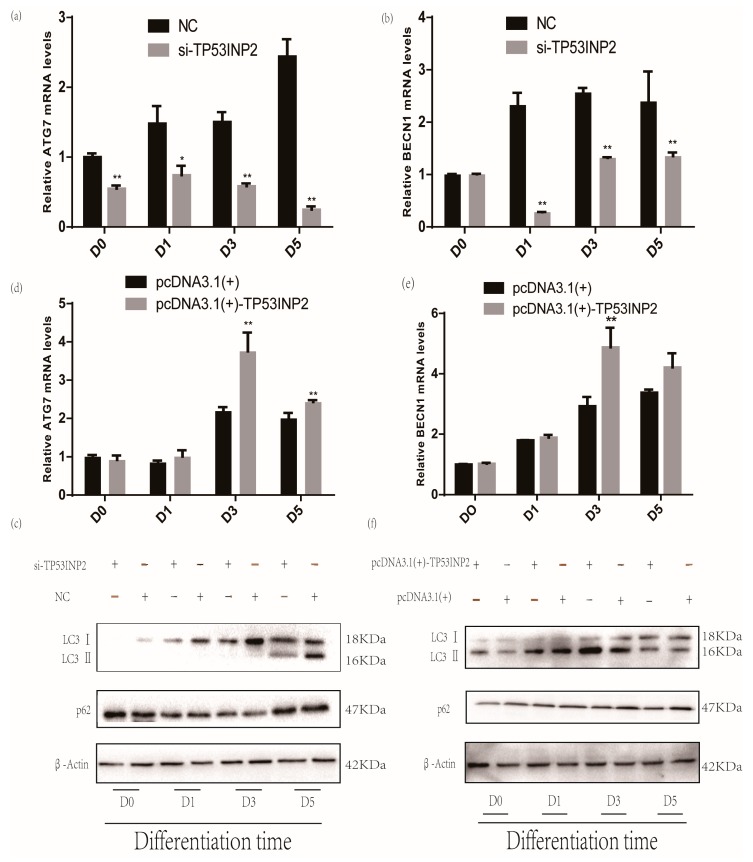
TP53INP2 induced autophagy. After interfering of TP53INP2, the mRNA expression of (**a**) *ATG7* and (**b**) *BECN1* were downregulated during adipocyte differentiation as detected by the RT-qPCR method. (**c**) LC3 and p62 protein levels during differentiation were evaluated by Western blot after interfering TP53INP2. After overexpressing of TP53INP2, the mRNA expression of (**d**) *ATG7* and (**e**) *BECN1* were upregulated during adipocyte differentiation as detected by the RT-qPCR method. (**f**) LC3 and p62 protein levels during differentiation were evaluated by Western blot after overexpressing of TP53INP2. Bar graphs are expressed as mean ± S. E.M. (*n* = 3), *, *p* < 0.05; **, *p* < 0.01, compared with NC and empty vector. NC, negative control.

**Figure 5 animals-09-01060-f005:**
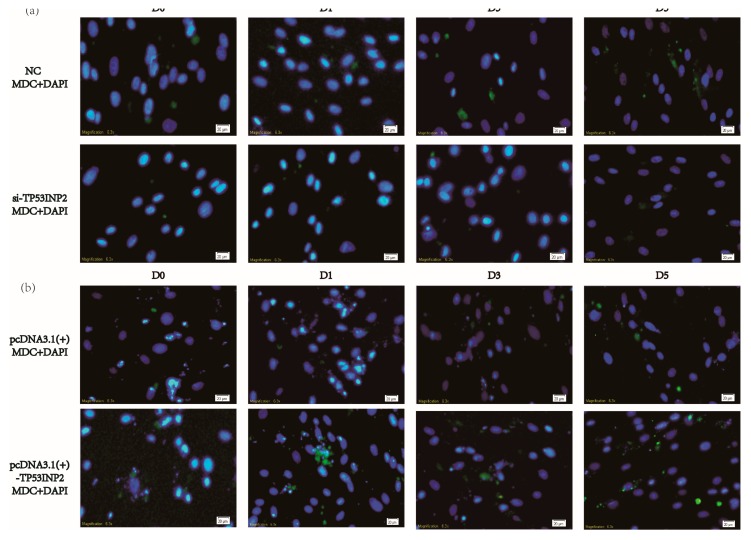
Dansylcadaverine staining to monitor autophagy flux in bovine adipocytes during differentiation. (**a**) Dansylcadaverine staining (green) used to monitor autophagy flux, the autophagy was inhibited after interfering TP53INP2 during adipocytes differentiation (20 μm). (**b**) Dansylcadaverine staining (green) used to monitor autophagy flux the autophagy was activated after overexpressing of TP53INP2 during adipocytes differentiation (20 μm). Compared with NC and empty vector. NC, negative control; DAPI, 4’,6-diamidino-2-phenylindole; MDC, Dansylcadaverine.

**Figure 6 animals-09-01060-f006:**
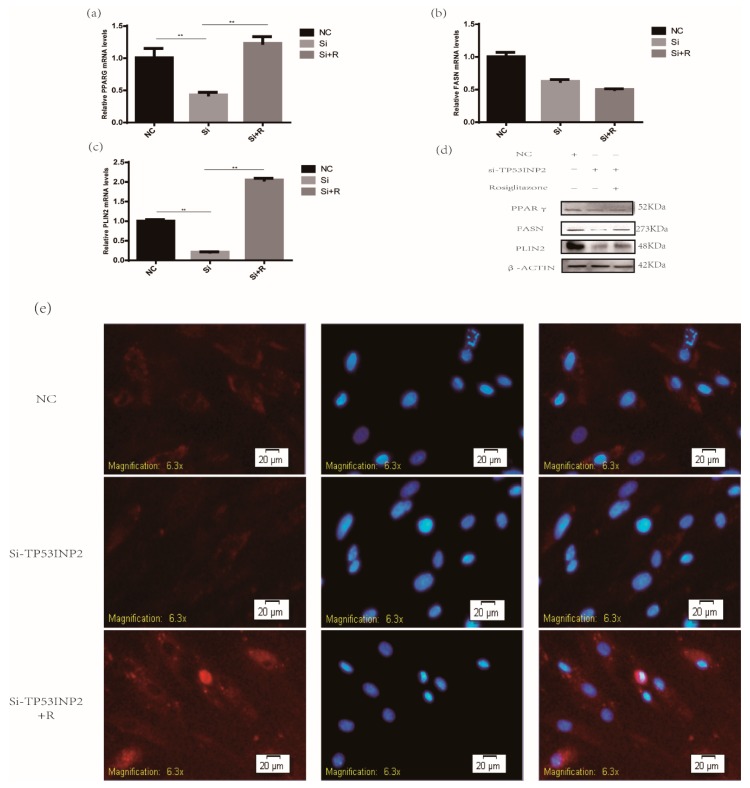
Activating the transcriptional activity of PPARγ can restore the decrease in lipid droplets and downregulation of differentiation genes caused by knocking down of TP53INP2. (**a**) *PPARγ*, (**b**) *PLIN2*, and (**c**) *FASN* mRNA expression were evaluated via RT-qPCR, rosiglitazone upregulated the mRNA levels of PLIN2 caused by decreasing the expression of TP53INP2. (**d**) PPARγ, PLIN2 and FASN protein expression were evaluated via Western blot, rosiglitazone upregulated the protein level of PLIN2 and FASN caused by decreasing the expression of TP53INP2. (**e**) Lipid droplets was stained via HCS LipidTOX™ Deep Red Neutral Lipid Stain (red) (20 μm), rosiglitazone restored the reduction of lipids caused by decreasing the expression of TP53INP2. Transfected TP53INP2 siRNA and NC into bovine adipocytes, next induced differentiation 3 days. Bar graphs are expressed as mean ± S.E.M. (*n* = 3), **, *p* < 0.01, compared with NC. NC, negative control; DAPI, 4’,6-diamidino-2-phenylindole; R, rosiglitazone.

**Table 1 animals-09-01060-t001:** List of the sequences of all primers, GenBank accession numbers and the size of the product in this study.

Gene	Primer	Sequence	Genebank Accession No	Product Size(bp)
*TP53INP2*	F	GCGGCTGTAGACTCAAAG	XM_003586843.5	130
	R	GTTATGAGGCGGAGTGTC		
*PPARγ*	F	GTTATGAGGCGGAGTGTC	NM_181024.2	117
	R	GTCCTCCGGAAGAAACCCTTG		
*PLIN2*	F	TGTCTACCAAGCTCTGCTC	NM_173980.2	210
	R	CGATGCTTCTCTTCCACTCC		
*FASN*	F	GATCCTCCTCATCCCAATAGTTC	NM_174662.2	117
	R	TTCAGTTGCCTCCCTTCATC		
*ATG7*	F	GCCAAAACAGATTCAAGCCCTCG	NM_001142967.1	109
	R	CAGCACCGTGGTCTCGTCATACTT		
*BECN1*	F	GATGTCCACAGAAAGTGCCAACA	NM_001033627.2	100
	R	GTCCCCAGTGACCTTTAGTCTTCG		
*β-Actin*	F	TCTAGGCGGACTGTTAGC	NM_173979.3	82
	R	CCATGCCAATCTCATCTCG		

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
