# Peer review of "TP53INP2 Promotes Bovine Adipocytes Differentiation Through Autophagy Activation"

_animals, 2019, doi:10.3390/ani9121060_

Round 1

Reviewer 1 Report

see attached

Reviewer 2 Report

This study investigates the role of TP53INP2 by using gain and loss-of-function studies in cattle adipocytes. The authors found TP53INP2 promotes adipocytes differentiation partly though Pparγ. In addition, the authors found that TP53INP2 induced autophagy during adipocytes differentiation. This is an interesting study which can be published in Animals after revision.

Major concern: The role of TP53INP2 in adipogenesis has been studied in vivo in model animal. In contrary to promoting adipogenesis in cattle adipocytes, TP53INP2 functions to inhibit adipogenesis in vivo in model animal. In this case, the data of this paper should be solid to draw the contrary conclusion. Generally, the quality of western blots and immunostaining should be improved and the results should be quantified.

Specifically,

Fig. 2e did not show the decrease of Pparγ at D1 and D3; FASN at D1 and D5; PLIN2 at D3 and D5. Fig. 2i did not show the increase of FASN at D1; PLIN2 at D1, D3 and D5. Fig. 2e and i need to be improved and quantified.

Fig. 3 The image quality is low, at least in the review pdf. The red color is too weak. Quantification of the imagines is required.

Fig. 4c p62 antibody did showed a discernable band for most of the samples. Fig. 4f did not show the increase of LC3 at D1 and D5, and did not show the increase of p62 at D0 and D1. Fig. 4c and f need to be improved and quantified. 

Fig. 5 The image quality is low and need to be improved. Quantification of the imagines is required. 

Round 2

Reviewer 2 Report

The authors have answered all my concerns. I recommend to publish this manuscript.